# The Mediating Effect of Age, Gender, and Post-Stroke Duration on the Association between Trunk and Upper Limb Recovery in Subacute Stroke Population: A Cross-Sectional Study with Mediation Analysis

**DOI:** 10.3390/ijerph192315644

**Published:** 2022-11-24

**Authors:** Kumar Gular, Viswanathan Sivasubramanian, Ravi Shankar Reddy, Jaya Shanker Tedla, Snehil Dixit

**Affiliations:** 1Division of Physical Medicine and Rehabilitation, Rajah Muthiah Medical College and Hospital, Annamalai University, Annamalai Nagar 608 002, India; 2Department of Medical Rehabilitation Sciences, College of Applied Medical Sciences, King Khalid University, Abha 61471, Saudi Arabia

**Keywords:** subacute stroke, upper limb impairment, hand functions, trunk control

## Abstract

Background: The trunk acts as proximal support with which limbs execute smooth and purposeful movement. Furthermore, as upper extremity functions are an integral component of daily living activities, exploring the association between trunk and upper extremity recovery will guide therapists in developing appropriate rehabilitation goals and interventions. The objectives of this study were to (1) assess the association between trunk and upper extremity recovery in the subacute stroke population and (2) assess the effect of trunk control on upper extremity impairment and function with age, gender, and duration of stroke as mediators using mediation analysis in subacute stroke individuals. Methods: This cross-sectional study included 54 subacute stroke participants with a mean age of 58.37 ± 6.11 years. The trunk impairment scale (TIS) assessed the trunk’s stability, mobility, and coordination. The level of upper extremity impairment was evaluated using the Fugl-Meyer Assessment scale (FMA). The quality and quantity of upper limb motor functions were measured using the Wolf motor function test (WMFT). Results: The TIS exhibited moderate positive correlations with the FMA-UE, WMFT-time scale (TS), and WMFT-functional ability scale (FAS) at *p* < 0.001. The mediation analysis reported a profound mediation effect of post-stroke duration on the association of trunk and upper limb recovery. Conclusions: The study results substantiated that trunk control significantly correlates with upper limb impairment and the quality and quantity of its use in the subacute stroke population. Post-stroke duration proved to mediate the association between trunk and upper limb recovery. Therefore, the assessment and intervention of trunk and upper extremity motor control considering the post-stroke duration is vital and should be incorporated in stroke rehabilitation aiming at functional independence.

## 1. Introduction

The significant increase in the incidence of stroke and its adverse impact on central nervous system functions are the foremost reasons for longstanding disability and mortality around the globe [1,2]. Ischemic strokes are the most common, followed by intracerebral and subarachnoid hemorrhages [3]. Stroke affects both genders, younger and older individuals, and lower socio-economic groups [4,5]. The early identification and modification of the risk factors and advanced stroke rehabilitation units will prevent first-time and recurrent strokes, respectively [6].

The paresis or paralysis of upper limb functions restricts self-care among the post-stroke population [7,8]. The recovery of upper extremity functions, especially dexterity, is hindered due to the deficiency of proximal and distal control during the acute stage of stroke [9,10]. Many daily living activities demand the usage of both hands, which are primarily affected due to the lack of participation of the affected upper extremity, which is a significant factor for long-term disability in the stroke population [11,12].

The trunk acts as proximal support with which limbs execute smooth and purposeful movement [13]. Antigravity muscle weakness, loss of trunk proprioception, and abnormal muscle tone on the affected side contribute to poor trunk control in stroke subjects [14,15]. Asymmetrical weight-bearing, lateral pelvic tilt of the pelvis on the side of hemiplegia, and loss of selective and sequential recruitment of trunk muscles impede the smooth execution center of pressure (COP) shift during upper limb reaching activities [16,17]. The ability of the upper limb to move freely in space for various functional activities requires anticipatory and reactive trunk postural adjustments [18,19]. Furthermore, these postural adjustments demand intact muscle activity from either side of the trunk [20].

In common, stroke causes paralysis of contralateral musculature, but as the trunk has bilateral representation, both contralateral and ipsilateral side trunk musculature will be affected [21]. Poor trunk posture and lack of stabilization of the scapula over the thorax affect the kinetics and kinematics of upper limb movement [22,23]. Even though there is the possibility of attaining upper limb functional recovery up to 6 months–1 year post-stroke, the maximum achievement of motor functions was quite evident between 2 weeks to 3 months [24]. Engaging and integrating affected upper extremities in rehabilitation are critical in attaining activities of daily living [25].

Trunk control and upper extremity functions are the core predictors for activities of daily living [26,27]. Trunk control proved to be an ideal independent predictor to elucidate ADL changes in post-stroke subjects [28]. In the early stages of stroke, trunk control established a moderate relation with the amount of performance of functional activities. In contrast, the level of upper extremity functions shows a lack of association with functional ability in the acute stages of stroke [29]. The application of external support to the trunk exhibited excellent scores on the trunk impairment scale (TIS) and upper extremity functions compared to the group without trunk support, which emphasizes the importance of a stable trunk in executing upper extremity functions in the subacute and chronic stroke population. Seok-Hui Yang et al. [30]. reported a significant improvement in upper limb motor functions and FMA-UE scores with *p* < 0.05 among the post-stroke subjects with a stable trunk [30]. Wee Sk et al. [31] found a significant improvement with trunk support on TIS scores (*p* < 0.001) and Streamlined Wolf motor function test (SWMFT) functional ability scores (*p* < 0.01) and time sores (*p* < 0.05) of upper extremity functions compared to without trunk support in chronic stroke patients [31]. The severe impairments and poor functional ability of the upper extremity presented a profound negative impact on prime components of functional independence, such as trunk control, balance, and mobility, in the subacute stroke population. 

Lee, K.B. et al. reported copious improvement in the trunk and upper extremity functions in the first three months and minor improvements between 3–6 months (Late subacute stage) of stroke [32]. The above results substantiated the moderate associations observed among subacute and chronic populations in the previous studies. However, proportionately, the amount of trunk control will reach its maximum compared to the slow recovery of upper limb motor functions in the first three months. The discrepancy in the motor improvements among the trunk and upper extremities in the first three months and the trivial improvements between 3–6 months questioned their relationship in the late subacute stage of stroke [32,33]. The insignificant improvements observed in the trunk and upper extremity motor functions between the successive months (3–4, 4–5, and 5–6) among the late subacute stroke population insisted on discovering the effect of the post-stroke duration (3rd, 4th, 5th, and 6th month) on their relationship. 

The non-modifiable factors, such as age and gender, were considered excellent indicators of functional recovery [34]. The younger the age of stroke onset, irrespective of the severity and associated complications of a stroke, the more reasonable the rate of recovery is, which tends to decline from the fifth decade onwards. Furthermore, regardless of the age of onset of the stroke, muscle strength and functional outcomes continue to improve until six months. Young people attained their maximum functions below three months post-stroke and had a constant improvement till 30 months. On the contrary, people ≥70 years showed a decline in functional improvement after one-month post-stroke [35,36]. The Structural and physiological changes due to ageing may slow the recovery process in post-stroke subjects compared to young people. The percentage of functional independence in females was relatively low compared to males of the same age group [37]. At three months post-stroke, regardless of the age and initial disability of the stroke, females showed poorer functional outcomes than males [38].

Independence in activities of daily living demands a stable trunk and less impaired and reasonable upper extremity motor control and their interrelations. The present study intended to investigate the relationship between trunk control and upper extremity function and explore the mediation effect of age, gender, and post-stroke duration in the late subacute stroke population, which will guide therapists to develop practical rehabilitation goals and interventions. Therefore, the objectives of this study were to (1) assess the relationship between trunk control and upper limb impairment and the quality and quantity of its use in the late subacute stroke population and (2) evaluate the mediation effect of age, gender, and post-stroke duration on the relationship between trunk control, upper extremity impairment, and function in late subacute stroke individuals. 

## 2. Materials and Methods

### 2.1. Study Design and Participants

The current cross-sectional study comprises post-stroke subjects referred to the King Khalid university physiotherapy clinic by a neurologist, neurosurgeon, or general physician from February 2020 to August 2021. The participants were 18–80 years old, had a first-time stroke, had a post-stroke duration of 3–6 months, and could follow commands. In addition, the post-stroke subjects excluded from the study were those suffering from other neurological, musculoskeletal, cardiorespiratory, perceptual, and visual disorders. 

The current research was scrutinized and accepted by the ethical research committee at King Khalid University with approval number (ECM #2021-4504)—(HAPO-06-B-001). Therefore, the current research obeyed the Declaration of Helsinki principles. Furthermore, all subjects who participated in the study provided their consent before starting. 

### 2.2. Sample Size Estimation

The sample size for the current cross-sectional study was computed considering the trend of the trunk impairment scale (TSI) score obtained from the pilot study (known population mean = 14.31, SD = 3.28, study group mean = 15.56), using a power of 0.80 and an alpha value of 0.05. Therefore, the estimated sample size was 54. G*power 3.1 (Universities of Dusseldorf, Dusseldorf, Germany) was utilized to estimate the sample size for one study group vs. population and continuous variable.

### 2.3. Data Collection and Measurement Tools

#### 2.3.1. Trunk Impairment Scale

The trunk impairment scale’s unique psychometric properties have proven it an ideal tool for evaluating trunk control in people suffering from a stroke [39,40]. The subcomponents of the TIS scale were static (3), dynamic (10), and coordination (4), for a total of 17 components. The maximum score for TIS was 23, of which 7 points were for the static subscale, 10 points for the dynamic subscale, and 6 points for coordination. Higher scores indicate reasonable trunk control. 

#### 2.3.2. The Fugl-Meyer Assessment of Upper Extremity

The Fugl-Meyer Assessment scale is an ideal tool to evaluate structural and functional impairment after stroke. The FMA scale encompasses sensory and motor functions, range of motion, coordination, balance, and reflex. The upper limb component of the Fugl-Meyer scale has 66 points with good reliability and consistency [41,42,43]. Therefore, the FMA scale was a widely accepted tool for evaluating the upper limb impairment level and motor recovery in post-stroke subjects. 

#### 2.3.3. Wolf Motor Function Test 

Wolf motor function test comprised of functional (15) and strength-related (2) components and assessments were performed in a series from proximal to distal joints and gross to fine motor functions of the upper limb. The time and quality of all 15 associated functional components of the upper extremity were measured. A blinded assessor evaluated the functional ability among 15 functional tasks on an ordinal scale with 6 points by observing the videotape recorded during the WMFT assessment schedule. The psychometric properties and the appropriate evaluation of the activity level of the upper extremity have convinced researchers worldwide to utilize it in several studies [43].

#### 2.3.4. Procedure

The included participants were briefed about the procedure after collecting written informed consent. Three investigators (K.G., V.K., and R.S.R.), with more than 15 years of experience in neurological physical therapy expertise in applying experimental tools, were involved in the data collection process. Demographic characteristics such as age, gender, height, weight, BMI, type of stroke, side of brain insult, affected side, hand dominance, post-stroke duration, and social status were collected on a data entry sheet. Height and weight were measured using a stadiometer and a weighing scale. Then, investigators administered the Trunk impairment scale (TIS), Fugl-Meyer Assessment scale for upper extremity (FMA-UE), and Wolf motor function test (WMFT), providing sufficient rest period. The TIS assessment was carried out with the participants comfortably sitting on a wide adjustable bed with thighs well supported on the surface, feet flat on the floor, knees at 90 degrees of flexion, trunk and head in midline, arm and hand placed over the thigh, and without trunk and arm support. The best of the three trials was considered for each item. The tests were verbally explained or demonstrated, and participants were not allowed any practice sessions. However, physical assistance was encouraged. The assessor followed the protocol given by Rehabilitation Medicine, University of Gothenburg [44], and the Wolf motor function test (WMFT) manual (Birmingham: University of Alabama, CI Therapy Research Group [45]) to administer FMA-UE and WFMT, respectively. 

### 2.4. Statistical Analysis

The dependent and independent variables in the study were assessed for normal distribution using the “Shapiro–Wilk Test”. Descriptive statistics were performed for all the variables and were represented as means and standard deviations (SD). To estimate the relationship between trunk control (TIS), level of upper limb structural and functional integration (FMA-UE), and time (WMFT-TS) and quality (WMFT-FS) of gross to fine motor functions of the upper limb, Pearson’s correlation coefficients statistical tool was performed. The r-value reflected insignificant correlation, feeble correlation, moderate correlation, convincing correlation, and very strong correlation if the values were (0.00–0.10), (0.10–0.39), (0.40–0.69), (0.70–0.89), and (0.90–1.00), respectively; *p* ≤ 0.05 was considered for all occasions. The mediation impact of age, gender, and post-stroke duration was computed using mediation analysis, which includes four steps (Figure 1).

Step 1: Using bivariate regression, estimate the total effect of an independent variable (TIS) on dependent variables (FMA-UE and WMFT-TS and WMFT-FAS). Step 2: estimate the direct effect (a) of TIS on age, gender, and post-stroke duration (3rd, 4th, 5th, and 6th month). Step 3a: estimate direct effect (c) of TIS on outcome variables (FMA-UE and WMFT-TS and WMFT-FAS) using multiple regression with TIS, age, gender, and post-stroke duration (3rd, 4th, 5th, and 6th month) as predictors and outcomes (FMA-UE and WMFT-TS and WMFT-FAS) as dependent variables. Step 3b: estimate the direct effect (b) of age, gender, and post-stroke duration (3rd, 4th, 5th, and 6th month) on outcome variables (FMA-UE and WMFT-TS and WMFT-FAS) using multiple regression with TIS, age, gender, and post-stroke duration as predictors and outcomes (FMA-UE and WMFT-TS and WMFT-FAS) as dependent variables. Step 4: Sobel test was performed to assess the indirect effect of age, gender, and post-stroke duration (3rd, 4th, 5th, and 6th month) on the association of TIS with FMA-UE and WMFT-TS and WMFT-FAS.

The significance value was accepted as *p* ≤ 0.05. IBM SPSS version 24.0 software (IBM Corp., Armonk, NY, USA) analyzed the study data.

## 3. Results

Fifty-four subjects fulfilled the current study’s selection criteria; their demographic characteristics, TIS, FMA-UE, and WMFT-TS and WMFT-FAS values are represented in Table 1.

Correlations between TIS, FMA-UE, and WMFT-TS and WMFT-FAS are detailed in Figure 2.

TIS exhibited moderate positive correlations with FMA-UE (r = 0.52), WMFT-time scale (TS) (r = −0.55), and WMFT-functional ability scale (FAS) (r = 0.50). In all cases, the *p*-value is less than 0.001

Mediation analysis using age, gender, and post-stroke duration as mediation: In step 1, trunk control showed a significant positive association with upper extremity impairment, FMA-UE (β = 3185, *p* =< 0.001), and upper extremity functions (WMFT-time scale (WMFTTS): β = −2.75, *p* =< 0.001 WMFT-functional ability scale (WMFTFAS): β = 0.220, *p* =< 0.001. Step 2 assessed the direct association of trunk control with age, gender, and post-stroke duration (3rd, 4th, 5th, and 6th month). Trunk control showed a significant relationship with post-stroke duration (B = 0.1.77, SE = 0.045, *p* = 0.001), and an insignificant relationship with age and gender at B = 0.301, SE = 0.479, *p* = 0.533 and B = 0.37, SE = 0.039, *p* = 0.324, respectively. In step 3a of the mediation analysis, TIS showed a significant direct effect on FMA-UE, WMFT TS, and WMFT FAS scores with *p* < 0.001. In step 3b of the mediation analysis, post-stroke duration (3rd, 4th, 5th, and 6th month) showed a significant mediation effect on the relationship between TIS and FMA-UE, WMFT TS, and WMFT FAS scores with *p* < 0.05. The direct effects of trunk control on upper extremity impairment and functions through the mediation of age, gender, and post-stroke duration are shown in Table 2.

The Sobel test (stage 4), which investigates the indirect effect (age, gender, and post-stroke duration as mediation) for statistical significance, is summarized in Table 3. The post-stroke duration had a mediation effect on upper extremity impairment (FMUE) and upper extremity function (WMFT FAS) with *p* < 0.05.

## 4. Discussion

The current study reveals the positive relationship of trunk control with upper limb impairment and function, which signifies that improved trunk control can lead to significant positive changes in upper extremity motor performance. Furthermore, the mediation analysis with age, gender, and post-stroke duration as mediators showed the considerable effect of post-stroke duration on the upper extremity quantity and quality of the amount of use in the late subacute stroke subjects (3–6 months).

The appreciable relation of trunk control with upper limb motor performance was justified by factors such as: 1. Requisition of trunk muscles anticipatory activation in providing appropriate postural stability and length–tension relationship of muscles around the trunk and shoulder for efficient upper extremity functions [46]; 2. Proactive postural control of the trunk allows ideal reaction time, movement duration, and functional range of motion in hand to accomplish the purposeful task [47]; and 3. The temporal and spatial coordination between the trunk and upper extremities is pivotal for efficient hand usage in self-care [48].

Aydoğan Arslan S et al. [33] reported a strong correlation between upper limb recovery, functional balance, and mobility with trunk stability, r = 0.49 to 0.80 at *p* < 0.001. In another study, Wee SK et al. [49] found strong correlations between TIS and Streamlined Wolf Motor Function Test time (SWMFT-Time), SWMFT-FAS, and FMA-UE. However, the reasons for the moderate positive correlations in the present study might be due to the participant’s duration of stroke being concise (4.80 ± 0.57 months), dampening of endogenic plasticity and functional improvements three months post-stroke, mean age of the participants, abnormal adaptive and compensatory movement schemes, and participant’s previous rehabilitation status [32].

In his longitudinal study, Lee et al. [26] reported less upper limb recovery than the lower leg and trunk control and a slow recovery process between 3 to 6 months. The stroke duration among the participants recruited for the present study was 4.80 ± 0.57 months, who scored 17.72 ± 1.76 on TIS and 28.33 ± 10.79 on Fugl-Meyer UE, emphasizing the persistence of mild to moderate impairment. Furthermore, the previous longitudinal study reported TIS scores at 19.07 ± 3.22 and FMA-UE at 29.75 ± 19.96.32 between 3–6 months post-stroke, which further reinforced the present study results. 

Previous research reports substantiated the moderate positive correlation of trunk control with upper extremity impairments (r = 0.52, *p* < 0.001), which emphasizes that execution of hand functions demands reasonable trunk stability and vice versa. Impaired trunk stability, asymmetrical trunk muscle activation, abnormal synergic patterns of the upper extremity, scapular instability, limitations in the upper extremity kinematics, and compensatory trunk movements hinder the balance and functional mobility in the late stages post-stroke [50,51]. 

The Pearson’s correlation results of the current study showed a positive association between trunk control (TIS) with upper extremity functions, WMFT-time scale (TS) (r = −0.55), and WMFT-functional ability scale (FAS) (r = 0.50) among subacute stroke population at *p* value < 0.001. Furthermore, the improvements in FMA-UE, ARAT, and WMFT trunk stabilization exercises prescribed on static and dynamic surfaces among post-stroke subjects represent the 3a step of the mediation analysis of the direct effects of trunk control on upper extremity functions [52]. 

In addition, the functional task approach engaging the affected upper extremities for reaching and grasping improves the TIS scores [53]. It emphasizes the positive associations between the trunk and upper extremity recovery reported in the present study. Rand et al. underscored the importance of temporal and spatial coordination between arm and trunk movements during various reaching movements. During extreme reaches, post-stroke subjects produce near-normal timing of trunk and arm muscle recruitment but fail to retain spatial coordination between arm and trunk, which leads to excessive trunk movements [53]. It demands the focus on trunk and upper extremity training simultaneously to improve the spatial coordination between the trunk and upper extremity to attain maximum reach. 

Trunk control showed a significant relationship with post-stroke duration (B = 0.1.77, SE = 0.479, *p* = 0.533), whereas there was an insignificant relationship with age and gender. Furthermore, the present study’s mediation analysis reported the significant effects of post-stroke duration on the positive association of trunk control with upper extremity functions with *p* < 0.05 in the late subacute stroke population, which infers that the impact of the trunk balance and coordination on upper extremity functional recovery may vary with the post-stroke duration. 

Impairments such as muscle weakness, spasticity, abnormal synergic patterns, and compromised sensations are predominant in the subacute stroke population, which may lead to learned nonuse or lousy use of the hand in independent functions. The limitations in the usage of the hand may further affect trunk stability [54,55]. The positive associations between the trunk and upper limb impairment observed in the present study suggested evaluating and minimizing the upper limb impairments and encouraging the utilization of the upper limb, which in turn can reduce the upper limb impairments and improve the trunk control, functional balance, and gait functions in late subacute stage [56,57].

The influence of post-stroke duration (3rd, 4th, 5th, and 6th month) on the relationship between the trunk and upper extremity recovery demands comprehensive evaluation and intervention of the trunk and upper extremity for better functional recovery in the late subacute stroke population. However, the mediation analysis results of the post-stroke duration of the present study on the association between the trunk and upper extremity conceded at a specific point with small sample size warrants considering the results cautiously.

The other mediator’s age and gender did not significantly influence the relationship between the trunk and upper extremity recovery. Even though elderly age is considered one of the risk factors for the occurrence of stroke, it cannot be the only factor to determine functional recovery in stroke, as both young and older adults improved significantly in their functions with specialized rehabilitation [36]. Apart from age and gender, other factors, such as motor impairments, memory disorders, perceptual disorders, sensory impairments, and biochemical imbalances, may affect functional recovery at various stages of stroke [35,58]. 

The current study is the first to report the mediation effect of age, gender, and post-stroke duration on the association of trunk control with upper limb impairment and quantity and quality of upper limb use at 3–6 months in post-stroke subjects. The above results will guide therapists to critically analyze and evaluate the trunk and upper extremity motor control and coordination level in identifying the problems and setting treatment strategies to improve the participation of the upper limb in day-to-day functional activities in the late subacute stroke population. However, there are a few limitations in the present study: 1. The mediation analysis considering post-stroke duration (3rd, 4th, 5th, and 6th month) showed that the duration of a stroke may affect the relationship of trunk control with upper extremity function but could not disclose the effect of each month; 2. The present study could not display the effect of the intervention on the trunk and upper extremity functions, as the measurement of their functional capacity was collected at a single point in time; 3.The sample size is considerably small to determine the mediation effect of age, gender, and post-stroke duration (3rd, 4th, 5th, and 6th month) on the trunk and upper extremity relations; 4. Lack of quantitative outcome measures; 5. No consideration of other potent mediators, such as Body mass index (BMI), type of the stroke, side of paralysis, hand dominance, cognitive level, perceptual disorders, previous rehabilitation status etc.; and 6. The current study could not rule out the effect of static stability, dynamic stability, and coordination of the trunk on upper extremity function.

In the future, a longitudinal study model with multiple time points (3rd, 4th, 5th, and 6th month) considering all the potent mediators and incorporating quantitative measurements can better understand the effect of post-stroke duration (3rd, 4th, 5th, and 6th month) on the trunk and upper extremity relations. Nevertheless, in the future, independent effects of static stability, dynamic stability, and coordination of the TIS scale on the upper extremity can be delineated to understand their impact better.

## 5. Conclusions 

The study results substantiated that trunk control significantly correlates with upper limb impairment and the quality and quantity of its use in the subacute stroke population. Post-stroke duration proved to mediate the association between the trunk and upper limb recovery. Therefore, the assessment and intervention of trunk and upper extremity motor control considering the post-stroke duration is vital and should be incorporated in stroke rehabilitation aiming at functional independence.

## Figures and Tables

**Figure 1 ijerph-19-15644-f001:**
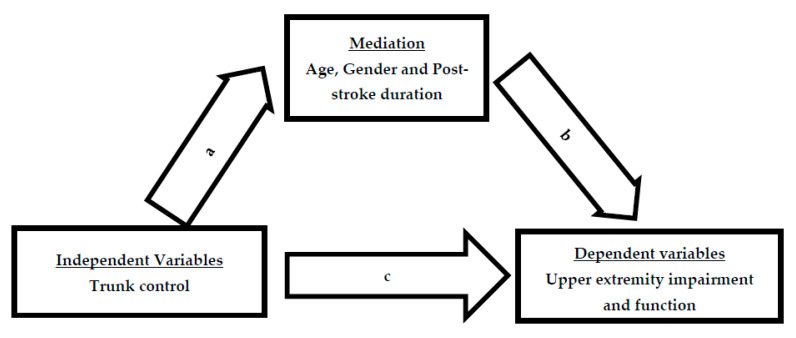
Mediation model including age, gender, and post-stroke duration as control variables.

**Figure 2 ijerph-19-15644-f002:**
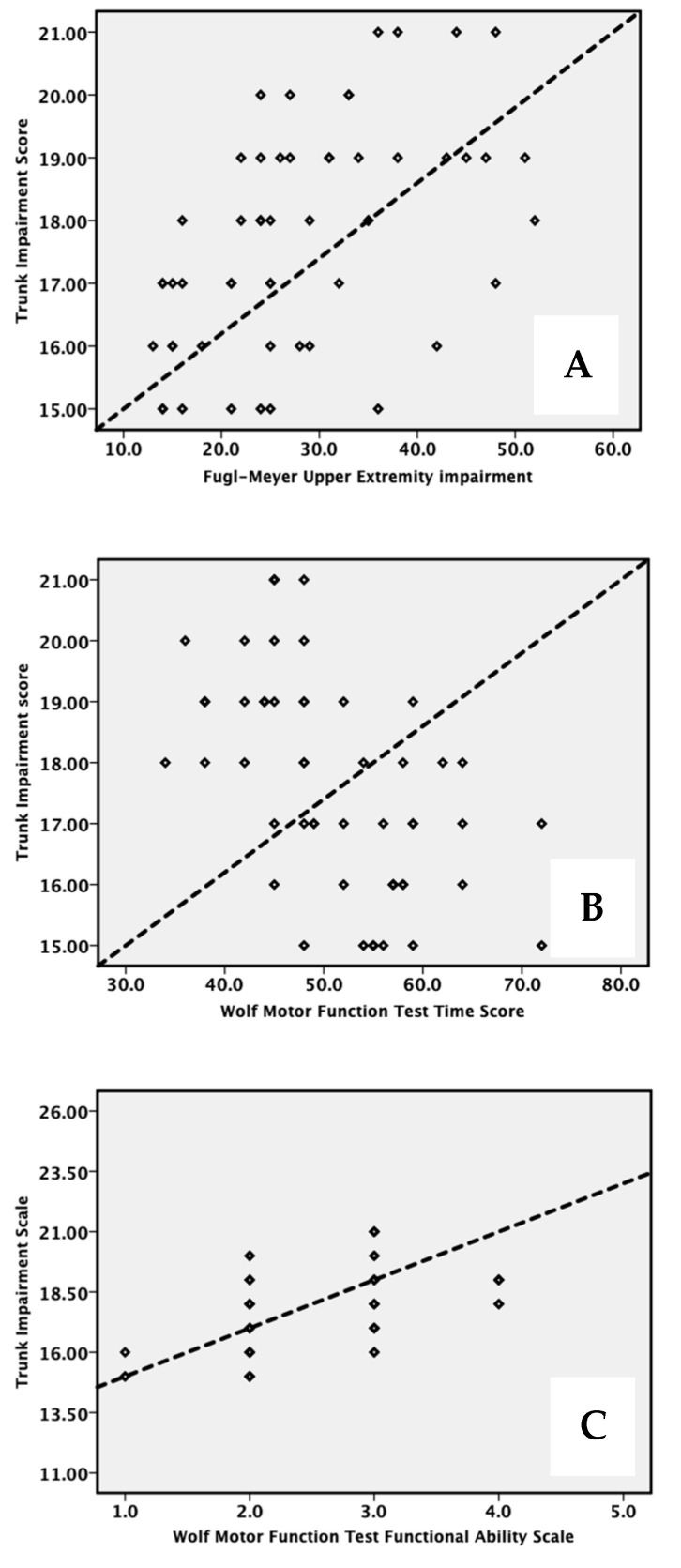
Correlation between trunk control (TIS) and (**A**) FMA-upper extremity, (**B**) WMFT-Time score and score, and (**C**) WMFT-Functional ability scale. Dashed line = regression line.

**Table 1 ijerph-19-15644-t001:** Characteristics of the study participants (*n* = 54).

Variables	Values
Age in years (Mean ± SD)	58.37 ± 6.11
Gender (Male/Female)	32/25
Height in meters (Mean ± SD)	1.66 ± 0.10
Weight in Kilograms (Mean ± SD)	72.05 ± 12.90
BMI (Mean ± SD)	26.42 ± 6.21
Type of stroke (Ischemic/Hemorrhagic)	44/10
Side of brain insult (Right/Left)	25/29
Side of paralysis (Right/Left)	29/25
Hand dominance (Right/Left)	52/2
Duration of stroke (Number of days)	144.25 ± 17.38
TIS score (Mean ± SD)	17.72 ± 1.76
FMUE score	28.33 ± 10.79
WMFT	
Time score (Mean ± SD)	50.77 ± 8.7
Functional ability score (Mean ± SD)	2.55 ± 0.76

TIS = Trunk Impairment Scale, FMUE = Fugl-Meyer-Upper extremity, WMFT = Wolf motor function Test.

**Table 2 ijerph-19-15644-t002:** Direct effects of trunk control on upper extremity impairment and functions with the effect of age, gender, and post-stroke duration.

Explanatory Variables	Direct Effect			Indirect Effect		
	B	SE	*p*-Value	B	SE	*p*-Value
Age × TIS × FMUE score	3.09	0.72	<0.001	0.29	0.20	0.167
Age × TIS × WMFT TS score	−2.62	−0.54	<0.001	−0.41	−0.15	0.011
Age × TIS × WMFT FAS score	0.21	0.05	<0.001	0.01	0.01	0.243
Gender × TIS × FMUE score	3.11	0.73	<0.0 01	−0.48	2.56	0.429
Gender × TIS × WMFT TS score	−2.63	0.56	<0.001	3.17	1.98	0.116
Gender × TIS × WMFT FAS score	0.21	0.53	<0.001	−0.04	0.18	0.822
PSD × TIS × FMUE score	2.17	0.77	<0.001	5.72	2.08	0.008
PSD × TIS × WMFT TS score	−2.18	0.63	<0.001	−3.19	1.7	0.006
PSD × TIS × WMFT FAS score	0.14	0.05	<0.001	0.41	0.15	0.009

TIS = Trunk Impairment Scale, FMUE = Fugl-Meyer-Upper extremity, WMFT = Wolf motor function Test, TS = Time score.

**Table 3 ijerph-19-15644-t003:** Sobel test for the indirect effect for statistical significance.

Explanatory Variables	Sobel-Test	SE	*p*-Value
Age × TIS × FMUE score	0.57	0.152	0.566
Age × TIS × WMFT TS score	0.61	0.203	0.540
Age × TIS × WMFT FAS score	0.62	0.081	0.530
Gender × TIS × FMUE score	0.18	0.947	0.851
Gender × TIS × WMFT TS score	1.51	0.742	0.114
Gender × TIS × WMFT FAS score	0.22	0.066	0.824
PSD × TIS × FMUE score	2.25	0.441	0.044
PSD × TIS × WMFT TS score	1.69	0.333	0.090
PSD × TIS × WMFT FAS score	2.24	0.32	0.024

TIS = Trunk Impairment Scale, FMUE = Fugl-Meyer-Upper extremity, WMFT = Wolf motor function Test, TS = Time score.

## Data Availability

The raw data for this study are uploaded in the Appendix A.

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
