# Peer review of "The Mediating Effect of Age, Gender, and Post-Stroke Duration on the Association between Trunk and Upper Limb Recovery in Subacute Stroke Population: A Cross-Sectional Study with Mediation Analysis"

_ijerph, 2022, doi:10.3390/ijerph192315644_

Round 1

Reviewer 1 Report

This paper studied assessing the association between the trunk and upper extremity recovery in the sub-acute stroke population and the effect of trunk control on upper extremity impairment and function through age, gender, and duration of stroke as mediators using mediation analysis in subacute stroke individuals. Because there is Few studies observed a steady improvement of upper extremity function and trunk control at multiple time points of post-stroke subjects ,this paper proposes that as upper extremity functions are an integral component of ADL, exploring the relationship and evaluating the mediation effect of trunk control on upper extremity functions

Major points

1. The significance of this paper is due to the background of neurological rehabilitation. Trunk control and upper extremity functions are the core predictors for activities of daily living. Trunk control proved to be an ideal independent predictor to elucidate ADL changes in post-stroke subjects. The demographics like age and gender were considered excellent indicators of functional recovery. And the interpretationsis very detailed and well-grounded. However the author did not summarize and review the research results of the predecessors, so that the reader did not understand well the latest progress of the current domestic and foreign studies, and did not have a good exposition of the theoretical and clinical results. Please sort out the latest research results and let us understand clearly.

2. The purpose of the cross-sectional study is specific with two scales and a function test. The article explains the scale and exercise test in detail, but does not explain the specific operation of the scale and the matters needing attention in the examination process, and the various screenings before participating in the trial, such as the basic physical condition and parameters of the subjects such as height, weight, gender, etc., that is, a basic and common physical examinations. What assumptions and premises are the experiments and content of the article based on? Whether it is possible to state some of the conditions and background that we should know more specifically, the author should add more details and attention. Another point to be mentioned is the experimental equipment and equipment and personnel training and preparation process, these details should be paid more attention. These are not elaborated in the article. For rehabilitation research, it is important to screen people before the experiment.

3. Where do the scientificity, rationality and reference of the experiment come from? Please come up with reliable reference sources and analyze the advantages and disadvantages of the experimental conditions, experimental methods and experimental results to reflect the characteristics of the system and compare with other methods proposed, which could make the experiment and results unconvincing.

4. This paper focuses on the process of statistical analysis and the analysis of the results. The statistical analysis process is clear and direct, but we should consider whether it makes full use of the data, and whether it can get more information we want and need through data statistical processing.

5. The results are detailed and clearly explained, and readers get what they want to know.The discussion section is sufficient, abundant, well-demonstrated, well-grounded, and persuasive, whether we have innovative and remarkable achievements and findings than others, pointing to mediation analysis, comparing with previous studies.

6. In terms of conclusions, that is positive and meaningful for us, but what are the implications for our current clinical significance and subsequent academic research? I hope the author can conduct in-depth discussion and analysis to explore. In addition, the limitations of this paper are not well explained, and what are the restrictive conditions and the parts that are not well controlled in the experiment process should be explained to the readers, so as to help the subsequent improvement. There is insufficient interpretation and literature support for our results. For example, why are there problems in the results of previous studies? Why can we improve and complete the problems that have not been solved by previous studies and get ideal results by avoiding those restrictive conditions. The authors could add the above explanations.

Minor points

1. The pictures are clear and the form is well explained, understandable and intelligible. For the chart section, the author has the ability to make things more beautiful and more perfect.

2. The introduction can be supplemented with a brief scientific study of the content and subject.

Author Response

Response to Reviewer comments

Thank you for your effort and time in reviewing our manuscript. The reviewing process has significantly improved the quality of this manuscript. Therefore, I am submitting this "Response to reviewers" document summarizing the changes we made in response to the critiques. In addition, I have highlighted the changes of manuscript in yellow color.

Reviewer 1

Major Concerns

Sl.no

Queries

Response to queries

1.      

The significance of this paper is due to the background of neurological rehabilitation. Trunk control and upper extremity functions are the core predictors for activities of daily living. Trunk control proved to be an ideal independent predictor to elucidate ADL changes in post-stroke subjects. The demographics like age and gender were considered excellent indicators of functional recovery. And the interpretations is very detailed and well-grounded. However, the author did not summarize and review the research results of the predecessors, so that the reader did not understand well the latest progress of the current domestic and foreign studies and did not have a good exposition of the theoretical and clinical results. Please sort out the latest research results and let us understand clearly.

Thanks for the valuable recommendations it helps us to convey the need of the study to the readers. The research findings from the forerunners were examined and described in accordance with the suggestions to help readers understand the need of present study

2.      

The purpose of the cross-sectional study is specific with two scales and a function test. The article explains the scale and exercise test in detail, but does not explain the specific operation of the scale and the matters needing attention in the examination process, and the various screenings before participating in the trial, such as the basic physical condition and parameters of the subjects such as height, weight, gender, etc., that is, a basic and common physical examinations. What assumptions and premises are the experiments and content of the article based on? Whether it is possible to state some of the conditions and background that we should know more specifically, the author should add more details and attention. Another point to be mentioned is the experimental equipment and equipment and personnel training and preparation process, these details should be paid more attention. These are not elaborated in the article. For rehabilitation research, it is important to screen people before the experiment.

The recommendations were considered and incorporated in the methodology section

3.      

Where do the scientificity, rationality and reference of the experiment come from? Please come up with reliable reference sources and analyze the advantages and disadvantages of the experimental conditions, experimental methods and experimental results to reflect the characteristics of the system and compare with other methods proposed, which could make the experiment and results unconvincing.

As per the recommendations Scientific, rationality and reference of the experiment was considered and incorporated  in the manuscript

4.      

This paper focuses on the process of statistical analysis and the analysis of the results. The statistical analysis process is clear and direct, but we should consider whether it makes full use of the data, and whether it can get more information we want and need through data statistical processing.

As per the objectives the data was utilized

5.      

The results are detailed and clearly explained, and readers get what they want to know. The discussion section is sufficient, abundant, well-demonstrated, well-grounded, and persuasive, whether we have innovative and remarkable achievements and findings than others, pointing to mediation analysis, comparing with previous studies.

Thanks for the comment

 We provided the implications of correlational and mediation analysis in detail in the discussion section of the manuscript

6.      

In terms of conclusions, that is positive and meaningful for us, but what are the implications for our current clinical significance and subsequent academic research? I hope the author can conduct in-depth discussion and analysis to explore. In addition, the limitations of this paper are not well explained, and what are the restrictive conditions and the parts that are not well controlled in the experiment process should be explained to the readers, so as to help the subsequent improvement. There is insufficient interpretation and literature support for our results. For example, why are there problems in the results of previous studies? Why can we improve and complete the problems that have not been solved by previous studies and get ideal results by avoiding those restrictive conditions. The authors could add the above explanations.

As per the recommendations In-depth discussion about the clinical significance of the results, limitations of the study and future scope of research was included in the discussion section

Minor concerns

1.      

The pictures are clear, and the form is well explained, understandable and intelligible. For the chart section, the author has the ability to make things more beautiful and more perfect

Thanks for the comments. The pictures were edited for better display

2.      

The introduction can be supplemented with a brief scientific study of the content and subject.

As per the recommendations scientific study of the content and subject. And their content were included

Reviewer 2 Report

Over view

I have ever read paper showing that trunk function affects the ability to stand up and walk (doi: 10.1371/journal.pone.0251977). On the other hand, while I had some sense that trunk function is naturally related to the ability to function the upper limbs and fingers, I had never encountered any studies that confirmed this scientifically and statistically, so I was interested to read this article. You even examined the effects of mediation, and I felt that you were conscientious to ascertain the true extent of the impact of trunk function on upper limb function.

Major concerns

1.     What are the reasons for choosing these three (age, gender, post-stroke duration) as mediation? Could other factors such as side of brain insult, paralysis on the dominant hand side, USN, somatagnosia, cognitive ability, etc. be considered as mediation?

2.     There was a moderate correlation between TIS and FMA-upper extremity, WMFT-Time score, and WMFT-Functional ability scale: Am I correct in assuming that the difference in post-stroke duration has an impact on the variation in FMA-upper extremity, WMFT-
Time score, and WMFT-Functional ability scale with respect to the TIS scores (as shown in Figure 2)?

3.     This time, it was shown that post-stroke duration may be a mediator of upper limb function, wasn’t there any correlation between post-stroke duration and upper extremity function?

4.     (Line 157-168 & 191-216) The use of the terms direct effect and indirect effect differs from previous paper (https://doi.org/10.1186/s12874-018-0654-z). Your explanation may be correct, but I would appreciate it if you could indicate the reference to which you referred. Let me further say, as the methods are divided into step3a and step3b, the results should also be divided in that way.

5.     (Line 257-268) The information mentioned here should be written in introduction part I think, as it is considered to be the impetus to perform this study.

6.     (Line 269-276) When upper extremities are moved, trunk must function, so when upper extremity training is performed, it can be said that trunk training is performed simultaneously. From this idea, shouldn't we insist that the results of reference [50] can be explained by the results presented in this study?

7.     (Line277-283) I think that this part is just repeating about the result.

8.     (Line 292-295) I think you should write this part as you have a plan to investigate with multiple time points in the future study.

9.     No references are listed after [36].

10.  (Line 297-300) What are you trying to say in this part? Are you trying to say that age is not a predictive indicator of functional recovery?

Minor concerns

1.     (Line 178-190) The details of Table 2 are written in line 188-190, so the table should be omitted.

2.     (Line 183-185) What do the dashed lines represent? Are they regression line? I think explanation is needed in the legend.

3.     (Line 159) At the part ”among TIS age, gender, and post-stroke duration”, I cannot make sense as sentence.

4.     (Line 196) It is written “(B=0.1.77, SE=0.479, p=0.533)”, is it significant?

5.     (Line 200-203 & line 208-212) I think these two parts are saying same things.

6.     (Line 209) Isn't “p>0.05” odd when you are saying it's significant?

7.     (Line 236) “p<0.00” is not right.

8.     (Line237, 238, 260) What does SWMFT stand for? You need explanation for the abbreviation.

9.     (Line 309-311) By " inability to display causal relationships", do you mean that you could not analyze by stroke type? By “lack of control groups”, do you mean that you did not perform analysis of healthy subjects?

10.  (Line 311-313) I think this part should also be written in the tone of the limitation argument.

Author Response

Response to Reviewer comments

Thank you for your effort and time in reviewing our manuscript. The reviewing process has significantly improved the quality of this manuscript. Therefore, I am submitting this "Response to reviewers" document summarizing the changes we made in response to the critiques. In addition, I have highlighted the changes in the manuscript in green color.

Reviewer 2

Major concerns

Sl.no

Queries

Response to queries

1.      

What are the reasons for choosing these three (age, gender, post-stroke duration) as mediation? Could other factors such as side of brain insult, paralysis on the dominant hand side, USN, somatagnosia, cognitive ability, etc. be considered as mediation?

.

In  the present study we considered only non-modifiable risk factors age and gender as the mediators in the present study. In future recommendations we emphasis to consider other potent factors like BMI, side of brain insult, paralysis on the dominant hand side, USN, somatagnosia, cognitive ability as mediators

2.      

There was a moderate correlation between TIS and FMA-upper extremity, WMFT-Time score, and WMFT-Functional ability scale: Am I correct in assuming that the difference in post-stroke duration has an impact on the variation in FMA-upper extremity, WMFT-
Time score, and WMFT-Functional ability scale with respect to the TIS scores (as shown in Figure 2)?

The depicted correlations between TIS and FMA-upper extremity, WMFT-Time score, and WMFT-Functional ability scale in figure 2 represents the associations between 3-6 months.

3.      

This time, it was shown that post-stroke duration may be a mediator of upper limb function, wasn’t there any correlation between post-stroke duration and upper extremity function?

For mediation analysis  divided the sample based on  post-stroke duration (3, 4, 5, and 6th months) and performed mediation analysis to find out whether 3, 4, 5, and 6th months of post stroke has any independent effect on the relationship between trunk and upper extremity motor functions. The mediation analysis showed that duration of stroke(3, 4, 5, and 6th months)  significantly mediated the relationship between TIS and FMA-upper extremity, WMFT-Time score, and WMFT-Functional ability scale. However couldn't disclose how much each month has effect of relationship between trunk and upper extremity motor functions. This is an limitation of the present study

In future a longitudinal study with multiple time point measures considering all the potent mediators and incorporating quantitative measurements can better understand post-stroke duration (3rd, 4th,5th, and 6th month) on the trunk and upper extremity relations.

4.      

(Line 157-168 & 191-216) The use of the terms direct effect and indirect effect differs from previous paper (https://doi.org/10.1186/s12874-018-0654-z). Your explanation may be correct, but I would appreciate it if you could indicate the reference to which you referred. Let me further say, as the methods are divided into step3a and step3b, the results should also be divided in that way.

As per the recommendations the results  step3a and step 3b were explained separately

5.      

(Line 257-268) The information mentioned here should be written in introduction part I think, as it is considered to be the impetus to perform this study.

Thanks for the recommendation, It helps us to substantiate the need  for conduction the study in the introduction part

6.      

(Line 269-276) When upper extremities are moved, trunk must function, so when upper extremity training is performed, it can be said that trunk training is performed simultaneously. From this idea, shouldn't we insist that the results of reference [50] can be explained by the results presented in this study?

Thanks for your valuable findings, we incorporated and discussed the importance of focusing trunk and upper extremity training simultaneously to improve the spatial coordination between trunk and upper extremity to attain maximum reach

7.      

(Line277-283) I think that this part is just repeating about the result.

As per the suggestion we removed the sentence

8.      

Line 292-295) I think you should write this part as you have a plan to investigate with multiple time points in the future study.

Thanks for the recommendations we added it in future recommendations

9.      

No references are listed after [36].

We added the references after 36 in the reference list

10.   

(Line 297-300) What are you trying to say in this part? Are you trying to say that age is not a predictive indicator of functional recovery?

Age is predictive indicator for functional recovery, but in the late sub-acute stage as per the results of mediation of analysis of present and past longitudinal studies  the functional improvements are equal in young and elderly people especially in long term rehabilitation

Minor concerns

1.      

(Line 178-190) The details of Table 2 are written in line 188-190, so the table should be omitted

Table 2 has been omitted as per the recommendations

2.      

(Line 183-185) What do the dashed lines represent? Are they regression line? I think explanation is needed in the legend.

As per the recommendation we explained the dotted line as regression line in the legend

3.      

(Line 159) At the part” among TIS age, gender, and post-stroke duration”, I cannot make sense as sentence.

It was corrected as “estimate the direct effect (a) of TIS on age, gender, and post-stroke duration”

4.      

  (Line 196) It is written “(B=0.1.77, SE=0.479, p=0.533)”, is it significant?

The values were mentioned wrong it was corrected in the manuscript and significant (B=0.1.77, SE=0.045, p=0.001)

5.      

 (Line 200-203 & line 208-212) I think these two parts are saying same things.

As per the recommendation it was corrected 

6.      

(Line 209) Isn't “p>0.05” odd when you are saying it's significant?

As per the recommendation it was corrected 

7.      

(Line 236) “p<0.00” is not right.

It is wrongly mentioned, the actual value is “p<0.001” the same was corrected in manuscript

8.      

(Line237, 238, 260) What does SWMFT stand for? You need explanation for the abbreviation.

SWMFT stands for Streamlined Wolf motor Function Test and the same was added in the manuscript

9.      

(Line 309-311) By " inability to display causal relationships", do you mean that you could not analyze by stroke type? By “lack of control groups”, do you mean that you did not perform analysis of healthy subjects?

We mean to tell that “The present study could not display the effect of the intervention on the relationships of the trunk and upper extremity functions as the trunk and upper extremity functions

 measured at a single point in time”

10.   

(Line 311-313) I think this part should also be written in the tone of the limitation argument.

We include the recommended part in limitations

Round 2

Reviewer 1 Report

The authors addressed all the concerns.

One minor comment: The value in sobel-test (second row in Table 3) should be 0.61.

Author Response

Response to Reviewer comments

Thank you for your effort and time in reviewing our manuscript. The reviewing process has significantly improved the quality of this manuscript. Therefore, I am submitting this "Response to reviewers" document summarizing the changes we made in response to the critiques. In addition, I have highlighted the changes in the manuscript in yellow color.

Reviewer 1

Major concerns

Sl.no

Queries

Response to queries

1.      

One minor comment: The value in sobel-test (second row in Table 3) should be 0.61.

The value in sobel-test in second row in Table 3 is now changed to 0.61.
